# Numerical Study on Effects of Wind Speed and Space Heights on Water Evaporating Performance of Water-Retained Bricks

**DOI:** 10.3390/e24111550

**Published:** 2022-10-28

**Authors:** Rubing Han, Zhimao Xu, Enshen Long

**Affiliations:** 1School of Civil Engineering and Architecture, Southwest University of Science and Technology, Mianyang 621010, China; 2College of Architecture and Environment, Sichuan University, Chengdu 610065, China; 3Sichuan Electric Power Design Consulting Co., Ltd., Chengdu 610016, China

**Keywords:** numerical study, evaporation performance, water-retained brick, wind speed, evaporation space height

## Abstract

Energy-saving roof renovation methods are effective ways to alleviate the urban heat island effect. In this paper, the authors propose three models of two-layer water-retained bricks, established the physical and mathematic models of the water-retained bricks, and then conducted a computational fluid dynamics (CFD) simulation on the effect of wind speed and evaporation space height on the water-evaporating performance of water-retained bricks. The results show that: (1) for the water-retained bricks with no-hole lids, macroscopic evaporation does not happen under the static wind conditions; with the increase of wind speed, the evaporating boundary layer thickness decreases, the water vapor concentration gradient in the boundary layer and the mass diffusion flux increase; (2) for the water-retained bricks with strip-hole lids, under the static wind condition, the evaporating performance of the water-retained bricks with strip-hole lids is better than that of bricks with no-hole lids; with the increase of wind speed, the evaporation of bricks with strip-hole lids is less affected by inlet airflow velocity than that of bricks with no-hole lids; (3) as for both the water-retained bricks with no-hole lids and with strip-hole lids, for a given wind speed, both the water vapor concentration gradient and the mass diffusion flux decrease as the evaporation space increases.

## 1. Introduction

With the increasing development of urbanization, the issue of urban environment quality is becoming one of the hot topics [1]. In summer, urban heat island effect is becoming significantly serious, and thus, urban livability is getting worse, which has seriously impacted people’s quality of life and physical health [2]. Meanwhile, the thermal environment of the top room of the existing building is poor, and the energy-saving reform for rooftop is becoming quite urgent than ever before [3,4]. Currently, the energy is increasingly tight and building energy consumption continues to increase; thus, it is of great significance to propose energy-saving roof renovation methods that can alleviate the urban heat island effect [5,6].

A water-retained rooftop is one of the effective energy-saving rooftop proposals. So far, many scholars have carried out relevant investigations to reduce urban heat island effect by using water-retained rooftops. Spanaki et al. [7] applied an aluminum layer of 0.15 m above free water surface in a pond to investigate a new evaporative cooling technique by adjusting the temperature of water and indoor air. The results showed that the evaporation loss has a significant impact on cooling efficiency and the indoor air temperature with roof cooling technology is 30% lower than indoor air temperature in buildings without roof cooling technology. Goudarzi and Mostafaeipour [8] compared four passive systems and found that the wind trap is the most effective refrigeration system in terms of energy efficiency. From the economic point of view, the water-retained roof and air collector are the most economical. Runsheng et al. [9] investigated a new evaporative cooling technique by constructing a pond covered with towels on a waterproof PVC sheet. The results show that the new evaporative passive roof cooling technology has better performance than the pool with removable insulation layer, the roof shadow pond with towel has the best cooling performance, and the covered pond has the worst cooling performance. Sabzi et al. [10] simulated three kinds of passive refrigeration systems: roof pool, water jacket and radiation shield. Compared with the experimental data of model houses built in Iran, the results show that a roof pool has the best refrigeration performance. Ahmad et al. [11] carried out an experimental study on rooftop pond architecture. The results show that the vapor released by this passive cooling technology can improve the system performance in summer but reduced the system performance in winter, so the vapor overflow should be controlled. Sharifi and Yamagata [12] systematically evaluated the setting conditions and performance of roof pool by referring to the literature. Through comparison, it is found that air temperature, water depth and roof material are the key factors affecting the performance of roof pool, which can achieve carbon neutrality and maintain indoor thermal comfort. Almodovar and Roche [13] determined the size of a roof pool system for a water–air heat exchanger. Compared with the thermal insulation roof, the battery cooling performance is better when the water–air heat exchanger is installed on the roof pool. Kharrufa and Adil [14] conducted an experimental study on the evaporative cooling system of the rooftop pool. The test results show that the system can further cool the compressor and reduce the cooling load by 88% compared to a normal room.

In addition, vast researchers conducted investigation on evaporating performance of water in a limited space. Sarma [15] theoretically analyzed the heat and mass transfer of condensing steam flowing in a turbulent pipe to the pipe wall. The numerical results of interfacial heat and mass balance equation and different system parameters are used to estimate the local average values of Nusselt number and Reynolds number of condensate. Moreover, Rao et al. [16] described the heat and mass transfer problem of high concentration non-condensing gas diffusing to the surface of the pipeline under the condition of laminar forced convection in vertical pipelines. The numerical results are consistent with existing experimental data. Jabrallah et al. [17] numerically and experimentally studied the heat and mass transfer of the water falling film evaporation in a closed rectangular cavity. The authors described the thermodynamic state of the heated film by means of the liquid temperature and evaporation flow rate. The exchanges at the liquid–gas interface are characterized in terms of local Sherwood and Nusselt numbers. Tang and Etzion [18] proposed one-dimensional mathematical model to investigate the cooling performance of a rooftop pond with movable insulation in a hot and dry climate. The model was developed based on the authors’ theory of water evaporation rates of wet and free water surfaces and surrounding air. The results show that the rooftop pond with movable insulation has a slightly worse performance as compared to that of roof pond with gunny bags floating on water surface. Pasquill [19] studied the evaporation from plane, free-liquid surfaces of relatively small dimensions into a tangential air stream and tested the rates of evaporation under the influence of a turbulent boundary layer. The results showed that the absolute rate of evaporation may be predicted correctly in order of magnitude. Vik and Reif [20] establish models of the evaporation from a thin liquid surface beneath a turbulent boundary layer and validate the model using wind tunnel experiments based on near-surface asymptotics of turbulent velocities and scalar fluctuations to study evaporation of nonbuoyant pollutants from a liquid surface beneath a turbulent boundary layer. Sun et al. [21] made the wall surface condense by lowering the wall temperature and then made the condensed water to evaporate by heating the wall to study the heat transfer process of condensation and evaporation in a two-dimensional cavity. The results showed that the transient flows during condensation and evaporation are very different. Feddaoui et al. [22] used the propulsion method to calculate numerical results for an air–water system to study the evaporative cooling process of liquid films in vertical pipes. The results showed that, when the liquid film cooling effect is better, the inlet liquid temperature is higher, the gas flow Reynolds number is higher, and the heat convection of the flowing water film is the main operation mode of interfacial heat dissipation. Sefiane et al. [23] examined two sets of data for the thermal capillary convection of organic liquids. The measurement results showed that the flow field of evaporated liquid is strongly affected by the existence of thermal capillary convection. Soma and Kunugi [24] measured the evaporation rates in glass gaps with different geometry to study the effect of meniscus perimeter and curvature on evaporation. It was found that the relationship between the capillary length and the characteristic length of the gap structure has different evaporation tendency. In addition, the proportional law between the evaporation rate and different interface shapes of the meniscus was deduced to predict the maximum evaporation rate, which is in a good agreement with the experimental data. Costa [25] numerically studied the time evolution process of the overall Nusselt number and Sherwood number on the container wall and the condensed water layer on the wall and disclosed the comprehensive heat and mass transfer process of the wet air in the container under the action of transient natural convection and revealed the flow structure in the envelope layer, as well as the changes of temperature and concentration with time. Soma and Kunugi [26] experimentally quantified the effect of curvature oppositions on meniscus evaporation between parallel glass plates on a purified water tank. The results showed that the surface curvature between plates increased with the decrease of the gap distance, and the evaporation rate and flux increased with the increase of curvature between the plates. By comparison, the evaporation rate and flux can be organized by curvature. Raimundo et al. [27] used wind tunnel measurements and 3D-CFD numerical simulations to investigate the relationship between the mean air thermal properties of evaporation from heated water surfaces and forced air flows. The results showed that the CFD program has a good ability to predict the phenomena involved. Moreover, the evaporation rate mainly depends on the air velocity. Water–air temperature difference and relative humidity also have important effects but much less than air velocity.

To the best knowledge of the authors, however, there is no study on the effects of wind speed and evaporation space height on water evaporating performance of water-retained brick as presented in Figure 1. In order to bridge the gap, the authors propose three models of water-retained bricks, and the following works will be carried out in this study:To establish the physical and mathematic models of the water-retained brick;To conduct CFD modelling and validation;To evaluate the effect of wind speed and evaporation space height on water evaporating performance of water-retained bricks with no-hole lids;To identify the effect of wind speed and evaporation space height on water evaporating performance of water-retained brick with strip-hole lids.

The novelty of this study is the first time to propose a practical theoretical and systematic measure to improve the roof energy-saving performance and, thus, to reduce urban heat island effect.

## 2. CFD Modelling

### 2.1. Physical Model of the Water-Retained Bricks

A kind of water-retained brick is proposed based on the cooling effect of passive evaporation of water. First of all, there is no tap water supply on the roof of the existing building, the water-retained brick should mainly collect and store rainwater. Secondly, the water in the water-retained brick could totally evaporate up. In order to guarantee the heat insulation effect after evaporation, other auxiliary heat insulation measures should be considered. Furthermore, the roofing after laying water-retained brick should maintain the function that can bear person. Finally, considering the convenience of transportation, the size and weight of the waterr-retained brick should not be too large. Extreme issues should be considered, such as how to drain water in the event of a sudden increase in rainfall, and how to reduce evaporation and maintain water depth in successive dry seasons, and whether the building can be recycled after demolition.

The water-retained brick is made of ceramsite, cement, river sand and water, and the mass ratio of them is 9:10:22:5; in addition, the density of the brick is 1200 kg/m^3^, the compressive strength is 5.0–25 MPa, and the thermal conductivity of the brick is 0.402–0.502 W/(m·K).

A schematic of a water-retained double-layer brick is shown in Figure 1. The brick is equipped with movable lid, which is divided into two types: one is without a strip-hole, and the other is with a strip-hole. The upper layer of the brick stores water, and the lower part is combined with the roof to form an air layer. Considering the drainage and heat insulation functions of the brick, the height of the air layer is set to 2 cm. Moreover, considering the building modulus, the transportation of water-retained bricks and the convenience of construction, the horizontal size of the water-retained bricks is set to 30 × 30 cm. A certain distance is maintained between the side wall of the cofferdam and the lid, which is conducive to ventilation and evaporation.

The simplified physical model of natural evaporation of a single water-retained brick is shown in Figure 2, and the thickness of the lid is 2 cm. Considering the strength of the lid, the distance between two strip-holes is set to 6 cm, and the length of the strip-holes is set at 18 cm, both ends of the strip-holes are semicircles.

According to actual application needs, 30 water-retained bricks are arranged side by side to form a nine-meter-long physical model of evaporation, as shown in Figure 3, when the numerical simulation is performed. Three kinds of evaporation models are: Model A (lid without strip-hole), Model B (lid with upright hole to the air flow; to be specific, the air flow direction is from left to right as shown in Figure 3b, and the direction of the hole is in the vertical direction of the air flow), and Model C (lid with parallel hole to the air flow; to be specific, the air flow direction is from left to right as shown in Figure 3c, and the direction of the hole is in the same direction of the air flow), respectively. The ventilation holes at both ends of each brick are connected to form a ventilation channel. The Δh in the figure is the distance from the upper edge of the cofferdam to the water surface. If Δh are set as 0 cm, 2 cm, 6 cm, and 10 cm, respectively, and the corresponding water depths are set as 12 cm (full of water), 10 cm, 6 cm, and 2 cm, respectively; then, the height of the evaporation space are generated as 6 cm, 8 cm, 12 cm, and 16 cm, respectively. The water surface of each brick is discontinuous with a length of 26 cm. Since the bricks are the same, there is no water vapor transferring between the bricks during the natural ventilation process; therefore, the four walls of the brick are set as adiabatic walls in the numerical simulation.

### 2.2. Mathematic Models

By ignoring the volume force and pressure gradient of the fluid and neglecting dissipative heat, chemical reactions, and energy transfer due to molecular diffusion, the following governing equations of heat and mass transfer of water surface evaporation of water-retained bricks are established.

As for the given models, the positions of the *x*, *y*, *z* axes are defined as described in Figure 2, and the mass transfer differential equation for incompressible fluid evaporation on the water surface:(1)$$\frac{D\rho_{A}}{D\tau }=D(\frac{\partial^{2}\rho_{A}}{\partial x^{2}}+\frac{\partial^{2}\rho_{A}}{\partial y^{2}}+\frac{\partial^{2}\rho_{A}}{\partial z^{2}})$$
where *ρ* is the density, *τ* is a certain time, and *x*, *y*, *z* are axes for the studied space.

Boundary conditions are set below:

Fixed concentration boundary condition: $\rho_{A}=\rho_{S}$

Wall boundary condition:$${\left.\frac{\partial \rho_{A}}{\partial x}\right|}_{x=0}=0$$
$${\left.\frac{\partial \rho_{A}}{\partial y}\right|}_{y=0}=0$$
$${\left.\frac{\partial \rho_{A}}{\partial z}\right|}_{z=0}=0$$

Fluid continuity equation for evaporation on the water surface:(2)$$\frac{\partial \rho }{\partial t}+\frac{\partial \rho u}{\partial x}+\frac{\partial \rho v}{\partial y}+\frac{\partial \rho w}{\partial z}=0$$
where *u*, *v*, *w* are the velocity vectors of the *x*, *y*, *z* axes.

Incompressible fluid momentum equations:(3)$$f_{x}-\frac{1}{\rho }\frac{\partial p}{\partial x}+\frac{\mu }{\rho }(\frac{\partial^{2}u}{\partial x^{2}}+\frac{\partial^{2}u}{\partial y^{2}}+\frac{\partial^{2}u}{\partial z^{2}})=\frac{\partial u}{\partial t}+u\frac{\partial u}{\partial x}+v\frac{\partial u}{\partial y}+w\frac{\partial u}{\partial z}$$
(4)$$f_{y}-\frac{1}{\rho }\frac{\partial p}{\partial y}+\frac{\mu }{\rho }(\frac{\partial^{2}v}{\partial x^{2}}+\frac{\partial^{2}v}{\partial y^{2}}+\frac{\partial^{2}v}{\partial z^{2}})=\frac{\partial v}{\partial t}+u\frac{\partial v}{\partial x}+v\frac{\partial v}{\partial y}+w\frac{\partial v}{\partial z}$$
(5)$$f_{z}-\frac{1}{\rho }\frac{\partial p}{\partial z}+\frac{\mu }{\rho }(\frac{\partial^{2}w}{\partial x^{2}}+\frac{\partial^{2}w}{\partial y^{2}}+\frac{\partial^{2}w}{\partial z^{2}})=\frac{\partial w}{\partial t}+u\frac{\partial w}{\partial x}+v\frac{\partial w}{\partial y}+w\frac{\partial w}{\partial z}$$

Boundary conditions are set below:

Entrance boundary condition: $u=u_{in}$;

Exit boundary condition: P=Pa;

The wall has no sliding boundary condition:$${\left.u\right|}_{x=0}=0$$
$${\left.u\right|}_{y=0}=0$$
$${\left.u\right|}_{z=0}=0$$

Heat transfer differential equation for evaporation on the water surface:(6)$$\frac{Dt}{D\tau }=a(\frac{\partial^{2}t}{\partial x^{2}}+\frac{\partial^{2}t}{\partial y^{2}}+\frac{\partial^{2}t}{\partial z^{2}})$$
where *t* is the temperature.

Boundary conditions are set below:

Constant temperature boundary condition:$${\left.t\right|}_{x=0}=t_{0}$$
$${\left.t\right|}_{y=0}=t_{0}$$
$${\left.t\right|}_{z=0}=t_{0}$$

Adiabatic boundary condition:$${\left.\frac{\partial t}{\partial x}\right|}_{x=0}=0$$
$${\left.\frac{\partial t}{\partial y}\right|}_{y=0}=0$$
$${\left.\frac{\partial t}{\partial z}\right|}_{z=0}=0$$

## 3. Numerical Method and Validation of CFD Model

### 3.1. Numerical Method

The 3D grid generator of Gambit is applied to mesh the water-retained brick model (Figure 4), and Fluent 19.0 is used to solve the governing equations. The nonlinear deviation governing equations are linearized by a series of algebraic equations. In the process of the simulation, a pressure-based, standard-order upwind scheme method is adopted. The SIMPLE algorithm is used to obtain the pressure field, and the standard k−ε is selected as turbulence model to predict a higher accuracy. The standard wall function is used near the wall. Moreover, water steam is used as a working fluid of which physical parameters are found in the NIST database. In the whole simulation process, the residual convergence limit for the equations is set to less than 10^−4^. One side of the air duct is set as the velocity inlet, and the other side is set as the pressure outlet; both the inlet and outlet are assumed to have a constant concentration of water vapor components, and the vapor diffusion at the inlet is considered. The water surface is set as a no-slip boundary, the water vapor component concentration is set to a constant value, and the relative humidity is set to 100%. The inlet and outlet of the air duct and the water surface are set as constant temperature boundaries, and the other walls are all adiabatic surfaces. The water surface temperature is set to the air wet bulb temperature of 27 °C, and the inlet and outlet of the air duct are set to the ambient air temperature of 32 °C and the relative humidity of 60%.

### 3.2. Validation of CFD Model

In order to verify the CFD model, the mass diffusion flux in each water-retained brick of Model A under different evaporation space heights and wind speeds are measured; that is, the net evaporation water depth of each brick in a certain period of time under different evaporation space heights and wind speeds are tested. The results show that the difference between the simulation results and the experimental data has a maximum of 12.9%, which implies that the CFD results are very close to the experimental data. Therefore, the CFD model can be used in the following numerical simulations.

## 4. Results and Discussion

### 4.1. Effects of Wind Speed and Evaporation Space Height on Water Evaporating Performance of Brick Lids without Strip-Holes

(1)The height of the evaporation space of 6 cm

Table 1 shows the different water vapor concentrations under different wind speeds when the evaporation space height of the brick with no strip-hole lid (Model A) is 6 cm. When the wind speed is 0 m/s, the water vapor concentration gradient at the inlet and outlet of the flow channel is large, and the mass diffusion flux is large, while the flow channel is full of saturated water vapor, and the mass diffusion flux is smaller than that at the flow channel inlet and outlet. When the wind speed increases to 0.2 m/s, the water vapor concentration gradient on the inlet water surface is larger than that in the static wind, and the evaporating boundary layer at about 180 cm from the inlet is developed to fill the flow channel. At the same time, it can be seen that the wind speed of 0.2 m/s disturbs the diffusion state at the outlet, and the water vapor concentration distribution changes; the water vapor concentration gradient at the outlet becomes smaller, and the evaporation weakens. The water vapor concentration along the cofferdam edge of the water-retained brick is lower than the water surface. As the distance from the inlet increases, the influence of the thickness of the upper edge of the cofferdam edge gradually decreases. Saturated water vapor begins to appear on the upper edge of the cofferdam wall, and the upper edge of the cofferdam wall is saturated with water vapor. When the wind speed is 0.8 m/s, the water vapor concentration gradient on the inlet water surface is the largest, and then the evaporating boundary layer gradually develops, and an evaporating boundary layer appears at the outlet; the water vapor concentration gradient on the water surface over the entire flow channel gradually decreases along the wind speed direction. Meanwhile, the influence of the thickness of the upper edge of the cofferdam wall of the water-retained bricks becomes smaller. When the wind speed is greater than 0.8 m/s, as the wind speed increases, and the thickness of the evaporating boundary layer decreases; in addition, the water vapor concentration gradient in the evaporating boundary layer increases, and thus, the mass diffusion flux increases.

(2)The height of the evaporation space of 8 cm

Table 2 displays the different water vapor concentrations under different wind speeds when the evaporation space height of the brick with no strip-hole lid (Model A) is 8 cm. When the wind speed is 0, the evaporative boundary layer appears at the inlet and outlet of the flow channel, the middle part maintains the evaporation saturation state, and the water vapor concentration gradient on the water surface at the inlet and outlet of the flow channel is larger than that in the middle part. When the wind speed is 0.2 m/s, the evaporating boundary layer with 270 cm from the inlet is developed to fill the flow channel, and the water vapor concentration gradient formed in the static wind state at the outlet is broken. When the wind speed is 0.4 m/s, the evaporating boundary layer at 390 cm from the inlet is developed to fill the flow channel. As the wind speed increases, the distance of the evaporating boundary layer from the inlet that can be developed to fill the flow channel increases, and the water vapor concentration gradient on the inlet water surface increases. When the wind speed increases to 2 m/s, the thickness of the outlet boundary layer is two-thirds of the outlet height. The evaporating boundary layer on the flow channel is distributed obliquely upward, and the water vapor concentration gradient on the water surface gradually decreases along the airflow direction. Compared with the evaporation space height of 6 cm, the corresponding water vapor concentration gradient on the water surface becomes smaller, the water vapor mass diffusion flux becomes smaller, and the distance required for the evaporating boundary layer to develop to fill the flow channel becomes shorter.

(3)The height of the evaporation space of 12 cm

Table 3 is the water vapor concentration distribution with the evaporation space height of 12 cm. As shown in the table, when the wind speed is 0.2 m/s, the evaporating boundary layer of 210 cm from the inlet can be developed to fill the flow channel. As the wind speed increases, the water vapor concentration gradient on the water surface increases. When the wind speed increases to 2 m/s, the evaporating boundary layer of 270 cm from the inlet is developed to the entire flow channel, the water vapor concentration gradient in the evaporating boundary layer is large, and the mass diffusion flux is large, while the water vapor concentration gradient in the middle part and the outlet are smaller than that at the inlet. Compared with the evaporation space height of 8 cm, the corresponding water vapor concentration gradient on the water surface becomes smaller, the water vapor mass diffusion flux becomes smaller, and the distance required for the evaporating boundary layer to develop to fill the flow channel becomes shorter.

(4)The height of the evaporation space of 16 cm

Table 4 shows the water vapor concentration distribution of the evaporation space height of 16 m. As illustrated in the table, when the wind speed is 0.2 m/s, and the evaporating boundary layer of 240 cm from the inlet is developed to fill the flow channel. When the wind speed is 2 m/s, the evaporating boundary layer of 340 cm from the inlet is developed to fill the flow channel. With the increase of wind speed, both the water vapor concentration gradient and the average water vapor concentration gradient at the inlet become larger, the thickness of the evaporating boundary layer becomes smaller, and the mass diffusion flux increases. Compared with the evaporation space height of 12 cm, the corresponding water vapor concentration gradient on the water surface changes less, so the water vapor mass diffusion flux changes less, and the distance required for the evaporating boundary layer to develop to fill the flow channel is very small.

To summarize, for Model A, when the wind speed is given, the water vapor concentration gradient on the water surface is the largest when the evaporation space height is 6 cm, and the water vapor mass diffusion flux is the largest. For a given evaporation space height, the larger the wind speed, the smaller the thickness of the evaporating boundary layer, the larger the water vapor concentration gradient on the water surface, and the larger the mass diffusion flux. That is to say, the smaller the wind speed, the larger the thickness of the evaporating boundary layer, the smaller the water vapor concentration gradient on the water surface, and the smaller the mass diffusion flux. Therefore, for a given evaporation space height, if the lid is opened with strip-holes, the water vapor can be discharged in time; that is, the water vapor can also be discharged from the outlet of the flow channel when the wind speed is low; thus, the mass diffusion flux can be increased. Therefore, the evaporation characteristics of the water-retained bricks with strip-holes in the lid at different wind speeds and evaporation space heights will be discussed in the next section.

### 4.2. Effects of Wind Speed and Evaporation Space Height on Water Evaporating Performance of Brick Lids with Strip-Holes

(1)The height of the evaporation space is 6 cm

Table 5 is a table of the water vapor concentration distributions of Model B in the evaporation space height of 6 cm. Figure 5 shows the local magnification of inlet water vapor concentration distribution under static wind conditions. As can be seen from the figure, obvious evaporating boundary layer is formed in the inlet and outlet of the passage, and the gradient of the inlet is the largest. This indicates that the inlet is in contact with the air under static wind condition, and the partial water vapor pressure in the air is relatively low, water vapor is easy to spread into the air, and the internal water vapor can only spread out through the holes of the lid, so the water vapor concentration gradient is smaller accordingly. In addition, it can be seen that the water vapor concentration gradient of the water-retained brick with strip-hole lids is larger than that of the water-retained bricks with no-hole lids. Therefore, the evaporating performance of the water-retained brick with strip-hole lids (Model B) is better than that of the water-retained bricks with no-hole lids under static wind conditions.

When the inlet airflow velocity increases to 0.2 m/s, it can be seen from Table 5 that the evaporating boundary layer about 140 cm from the inlet can be developed to fill the flow passage. The evaporating boundary layer thickness gradually increases along the direction of wind speed, and the water vapor concentration gradient gradually decreases.

The distance of the evaporating boundary layer to fill the flow passage increases with the increase of the airflow velocity. When the inlet flow velocity is 0.8 m/s, the vapor concentration boundary layer develops to fill the flow passage at the point of 350 cm from the inlet. However, when the inlet flow velocity of the bricks with no-hole lids is 0.8 m/s, the evaporating boundary layer has already filled the flow passage. Even if the wind speed increases to 2 m/s, the evaporating boundary layer of the brick with strip-hole lids filling flow channel is only 530 cm from the inlet. It can be seen that, when the wind speed increases from 0.2 m/s to 2 m/s, under the same working conditions, the average concentration gradient of brick with strip-hole lids is smaller than that of bricks with no-hole lids. Thus, the evaporation of brick with strip-hole lids is less affected by inlet airflow velocity than that of bricks with no-hole lids.

Table 6 is a summary of the water vapor concentration distribution of Model C in the evaporation space height of 6 cm. Figure 6 shows the local magnification of inlet water vapor concentration distribution of Model C under static wind conditions. As can be seen from Figure 6, the water vapor concentration gradient of the brick at the entrance of the channel is relatively large, and the influence range of outlet and inlet conditions is basically consistent with that of Model B. In terms of the concentration gradient of the captured position, the concentration gradient in Model C is smaller than that in Model B, but the water vapor concentration at the strip-hole is the same in both models. In addition, it can be seen from Table 6 that, when the wind speed increases, the water vapor concentration gradient on the inlet water surface increases, which is similar to Model B. Comparatively speaking, the evaporating performance of Model C is slightly worse than that of Model B. Therefore, only the simulation results of Model B are presented below.

(2)The height of the evaporation space is 8 cm

Table 7 is a summary of the water vapor concentration distribution of Model B at the height of 8 cm in the evaporation space. As can be seen from Table 7, under the static wind conditions, the water vapor concentration gradient at the inlet and outlet of the flow passage is large, and the mass diffusion flux is large, while the water vapor concentration gradient of the central bricks is the same and smaller than that of the inlet and outlet. Therefore, the mass diffusion flux at both ends of the flow passage is greater than that at the middle under static wind conditions. When the wind speed increases, the evaporating boundary layer develops gradually. The greater the wind speed, the greater the distance between the point of the evaporating boundary layer filling the flow passage and the inlet. When the wind speed reaches 2 m/s, the evaporating boundary layer is 150 cm from the inlet that can fill the flow channel. Compared with the height of evaporation space of 6 cm, at the same wind speed, the distance between the point of evaporating boundary layer filling the flow passage and the inlet is smaller, the corresponding water surface water vapor concentration gradient is smaller, and the water vapor mass diffusion flux is smaller.

(3)The height of the evaporation space is 12 cm

Table 8 is a summary of the water vapor concentration distribution of Model B with a height of 12 cm in the evaporation space. As can be observed from Table 8, under the static wind conditions, the water surface in the two bricks at the inlet and outlet of the flow passage presents a large concentration gradient. The water vapor concentration gradient of the first brick at the outlet and inlet is larger than that of the second brick, and the water vapor concentration gradient in the middle bricks is the same and smaller than that of the inlet and outlet. As the wind speed increases, the evaporating boundary layer gradually develops backward along the direction of the wind speed. Compared with the evaporation space height of 8 cm, the distance between the point of evaporating the boundary layer developing to fill the flow passage, and the inlet is smaller, the average concentration gradient on the water surface is smaller, and the mass diffusion flux is smaller.

(4)The height of the evaporation space is 16 cm

Table 9 is a summary of the water vapor concentration distribution of Model B with evaporation space height of 16 cm. As can be revealed from Table 9, under the static wind conditions, the water vapor concentration gradient of the four water-retained bricks at the outlet and inlet of the flow passage is large, and the closer they are to the inlet and outlet, the larger the water vapor concentration gradient and the mass diffusion flux. The water vapor concentration gradient of the middle water-retained bricks is almost the same and smaller than that of the outlet and inlet. Along the direction of the wind speed, the gradient of water vapor concentration decreases gradually. The water vapor concentration gradient at the inlet is the largest, and the water vapor concentration gradient at the outlet is the smallest. With the increase of the wind speed, the average concentration gradient of the water vapor and the mass diffusion flux increases. Compared with the height of the evaporation space of 12 cm, the distance between the point of the evaporating boundary layer developing to fill the flow passage and the inlet is smaller, the average concentration gradient on the water surface is smaller, and the mass diffusion flux is smaller.

In conclusion, for the water-retained brick with strip-hole lids, under static wind conditions, the evaporating performance of the water-retained brick with strip-hole lids is better than that of the water-retained bricks with no-hole lids. With the increase of evaporation space height, the influence of external air environment on the surface evaporating performance of water-retained bricks at the inlet and outlet of flow passage is larger, and when the evaporation space height reaches 16 cm, the water vapor concentration gradient of four bricks at the inlet and outlet of the channel is larger. The outermost brick has the largest water vapor concentration gradient and the largest mass diffusion flux. With the increase of the wind speed, the height of the evaporation space increases, and the influence of the wind speed decreases. Although the concentration gradient of water vapor at the inlet is the same, the distance between the point of the evaporating boundary layer developing to fill the flow passage and the inlet becomes smaller, and the diffusion flux of water vapor becomes smaller. When there is a certain inlet wind speed, the evaporation of brick with strip-hole lids is less affected by inlet airflow velocity than that of bricks with no-hole lids.

## 5. Conclusions

In this work, the effects of wind speed and evaporation space height on the evaporating performance of water-retained bricks with no-hole lids and with strip-hole lids under three models were investigated by using numerical methods. The main findings are as follows:(1)For the water-retained bricks with no-hole lids, when the evaporation space height is given, the evaporation space is saturated with water vapor under the static wind condition, and macroscopic evaporation is no longer happened. With the increase of wind speed, water vapor can be discharged in time, and the evaporating boundary layer thickness decreases; thus, the water vapor concentration gradient in the boundary layer and the mass diffusion flux increase.(2)For the water-retained brick with strip-hole lids, when the evaporation space height is given, the water vapor concentration gradient at the inlet and outlet of the flow passage and the openings of the lids are larger under static wind conditions. With the increase of the wind speed, the distance between the point of the evaporating boundary layer developing to fill the flow passage and the inlet increases, and the influence range of the inlet air flow gradually increases. However, when there is a certain inlet wind speed, the evaporation of brick with strip-hole lids is less affected by the inlet airflow velocity than that of bricks with no-hole lids.(3)For both the water-retained bricks with no-hole lids and with strip-hole lids, when the wind speed is given, the evaporation space increases, the water vapor concentration gradient decreases, and the mass diffusion flux decreases.

The experimental research on the thermal insulation performance of the roof with water-retained bricks under different weather conditions, as well as different seasons, have not been investigated in the current study; thus, relevant studies will be carried out in the near future.

## Figures and Tables

**Figure 1 entropy-24-01550-f001:**
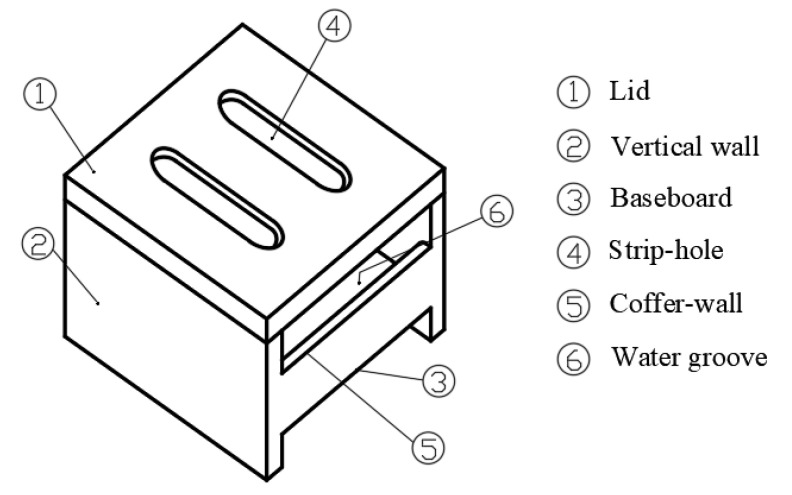
Schematic of a water-retained brick.

**Figure 2 entropy-24-01550-f002:**
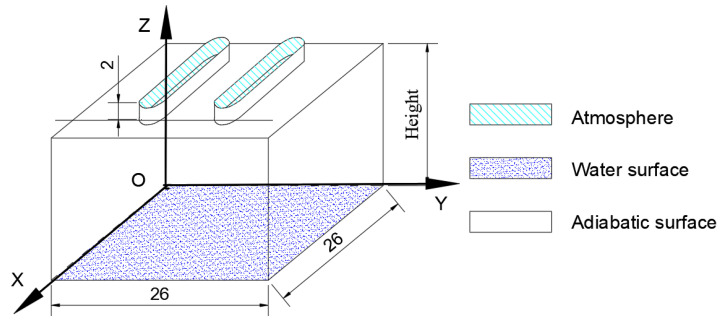
Evaporation model of a water-retained brick (Unit: cm).

**Figure 3 entropy-24-01550-f003:**
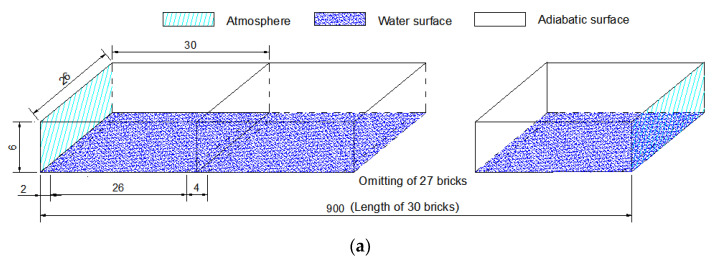
Evaporation model of water-retained bricks (Unit: cm). (**a**) Model A (Lid without a hole). (**b**) Model B (Lid with upright hole to the air flow). (**c**) Model C (Lid with parallel hole to the air flow).

**Figure 4 entropy-24-01550-f004:**
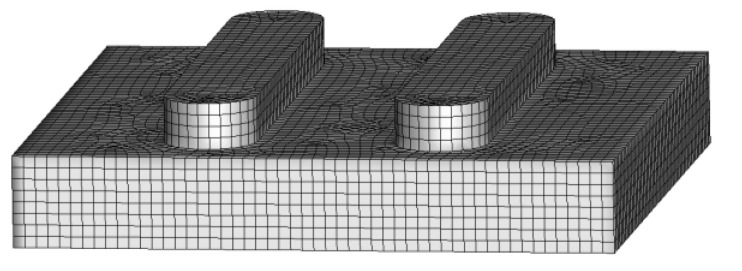
Meshing of one water-retained brick.

**Figure 5 entropy-24-01550-f005:**
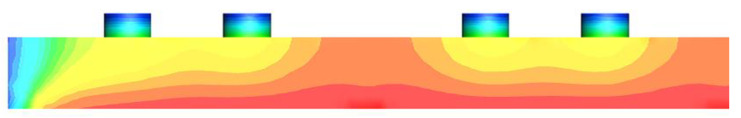
Local magnification of inlet water vapor concentration distribution in Model B under static wind conditions.

**Figure 6 entropy-24-01550-f006:**
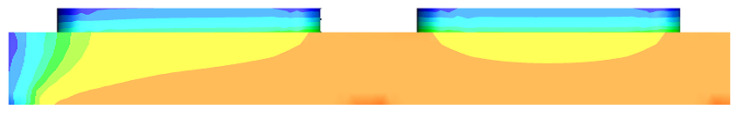
Local magnification of the inlet water vapor concentration distribution in Model C under static wind conditions.

**Table 1 entropy-24-01550-t001:** Water vapor concentrations under different wind speeds with an evaporation space height of 6 cm (Model A).

Wind Speed(m/s)	Water Vapor Concentrations 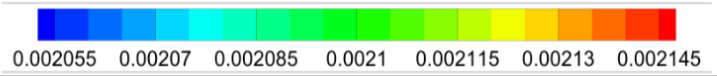
0	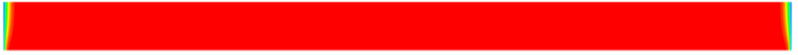
0.2	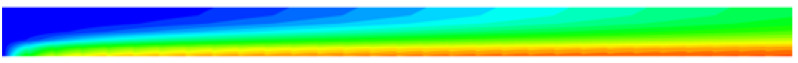
0.4	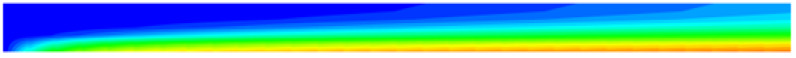
0.6	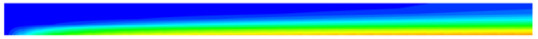
0.8	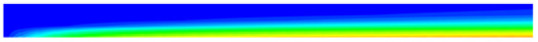
1.0	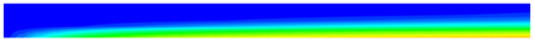
1.5	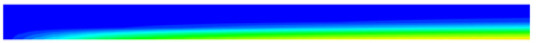
2.0	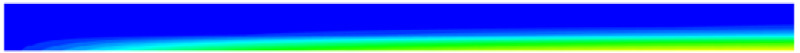

**Table 2 entropy-24-01550-t002:** Water vapor concentrations under different wind speeds with an evaporation space height of 8 cm (Model A).

Wind Speed(m/s)	Water Vapor Concentrations 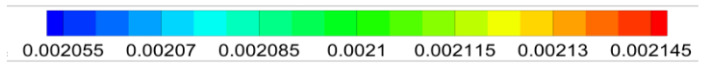
0	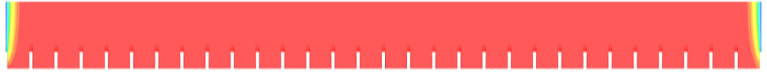
0.2	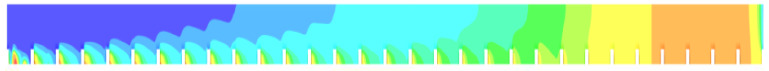
0.4	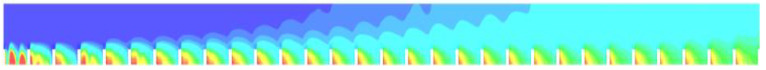
0.6	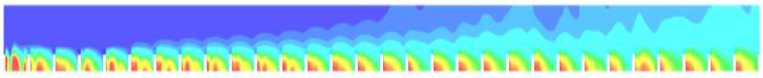
0.8	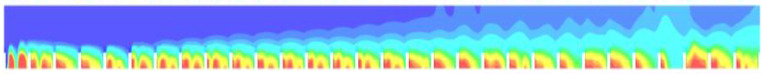
1.0	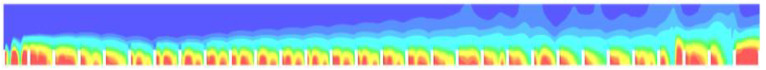
1.5	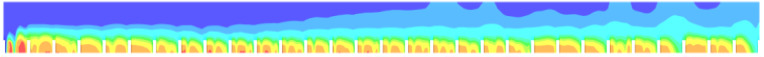
2.0	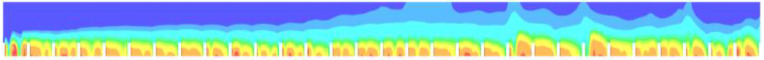

**Table 3 entropy-24-01550-t003:** Water vapor concentrations under different wind speeds with an evaporation space height of 12 cm (Model A).

Wind Speed(m/s)	Water Vapor Concentrations 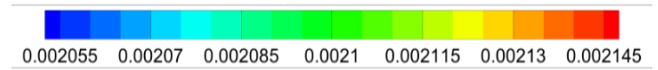
0	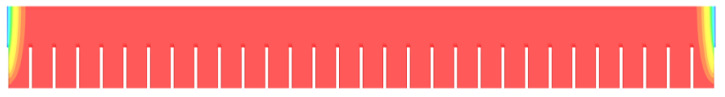
0.2	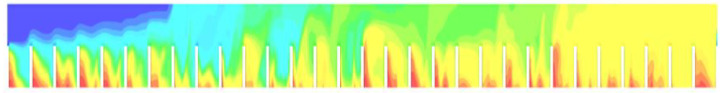
0.4	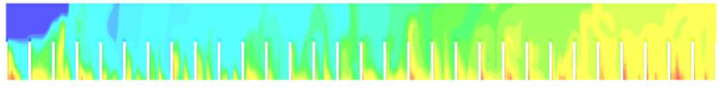
0.6	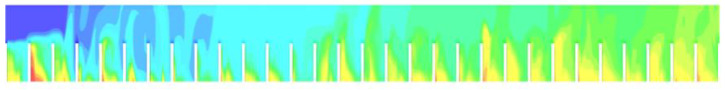
0.8	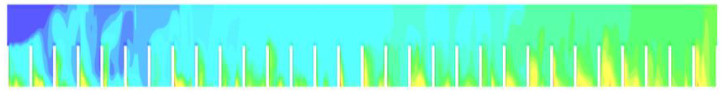
1.0	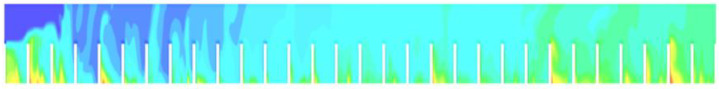
1.5	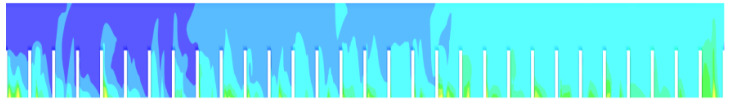
2.0	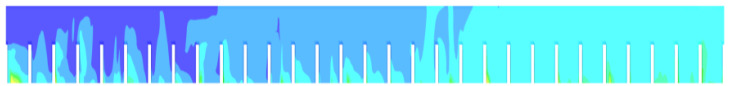

**Table 4 entropy-24-01550-t004:** Water vapor concentrations under different wind speeds with an evaporation space height of 16 cm (Model A).

Wind Speed(m/s)	Water Vapor Concentrations 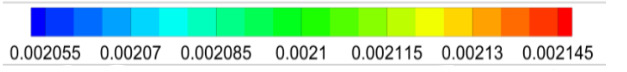
0	
0.2	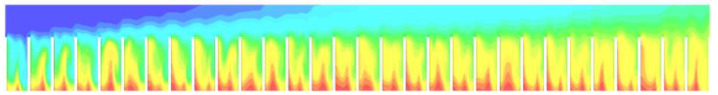
0.4	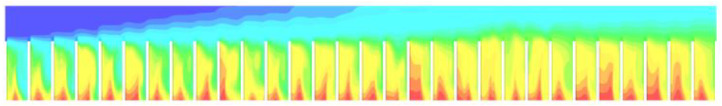
0.6	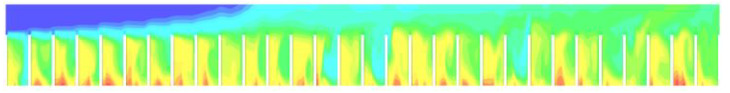
0.8	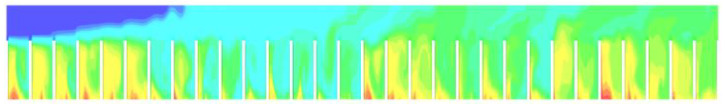
1.0	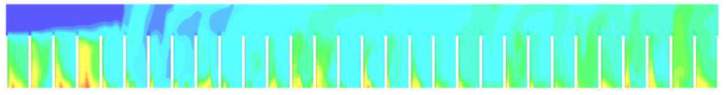
1.5	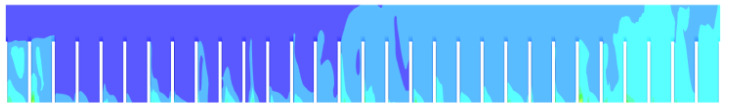
2.0	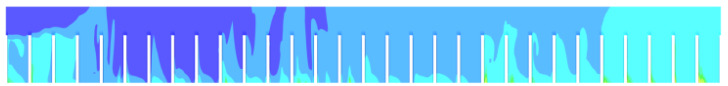

**Table 5 entropy-24-01550-t005:** Water vapor concentrations under different wind speeds with an evaporation space height of 6 mm (Model B).

Wind Speed(m/s)	Water Vapor Concentrations 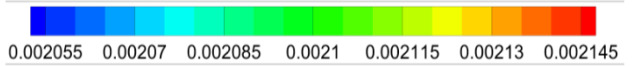
0	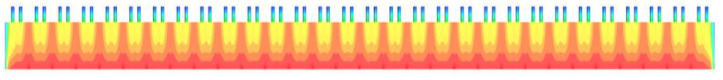
0.2	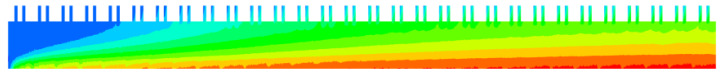
0.4	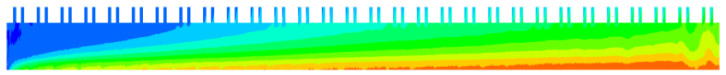
0.6	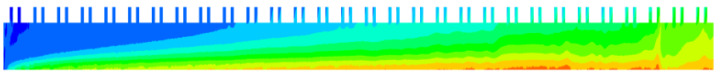
0.8	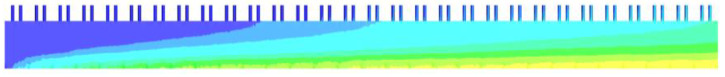
1.0	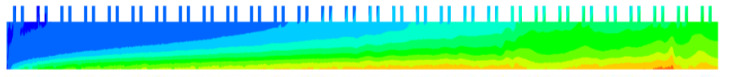
1.5	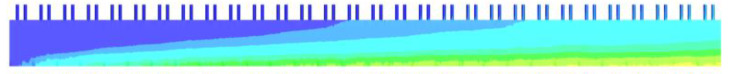
2.0	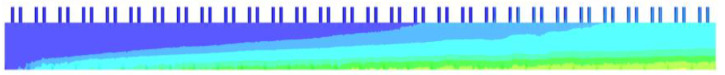

**Table 6 entropy-24-01550-t006:** Water vapor concentrations with an evaporation space height of 6 cm of the bricks (Model C).

Wind Speed(m/s)	Water Vapor Concentrations 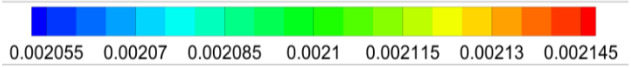
0	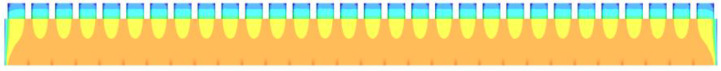
0.2	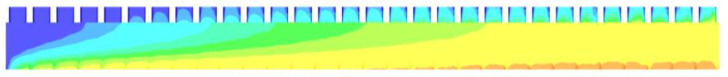
0.4	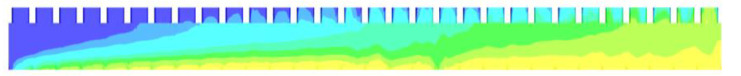
0.6	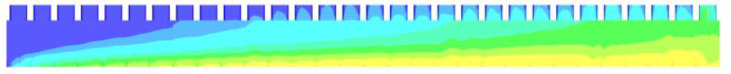
0.8	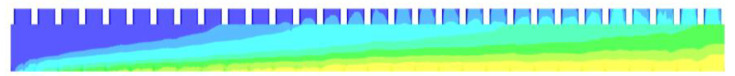
1.0	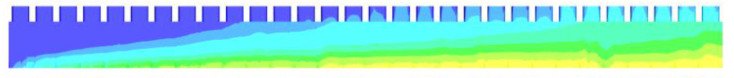
1.5	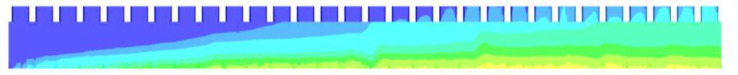
2.0	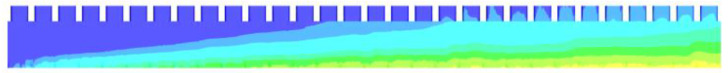

**Table 7 entropy-24-01550-t007:** Water vapor concentrations under different wind speeds with an evaporation space height of 8 mm (Model B).

Wind Speed(m/s)	Water Vapor Concentrations 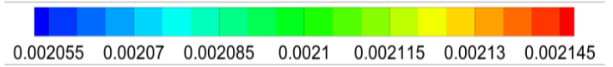
0	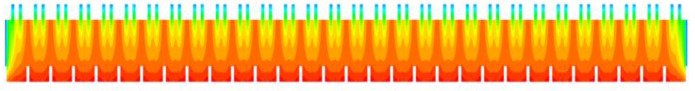
0.2	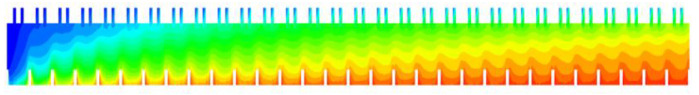
0.4	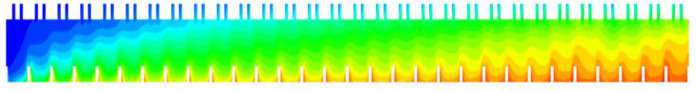
0.6	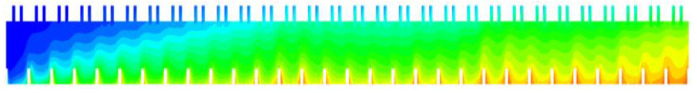
0.8	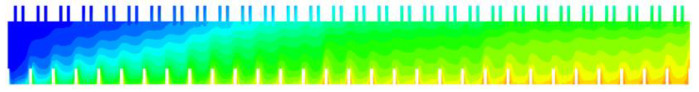
1.0	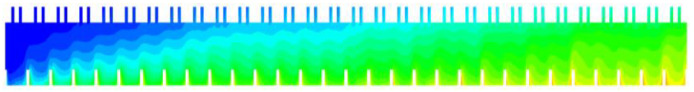
1.5	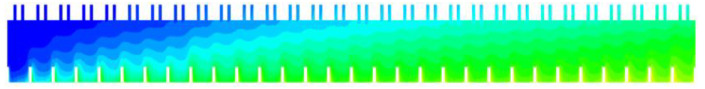
2.0	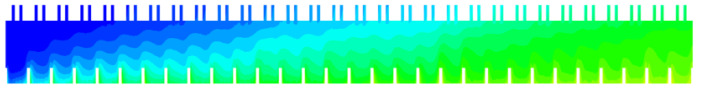

**Table 8 entropy-24-01550-t008:** Water vapor concentrations under different wind speeds with an evaporation space height of 12 mm (Model B).

Wind Speed(m/s)	Water Vapor Concentrations 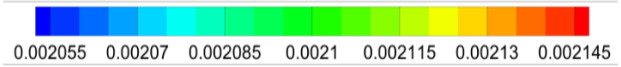
0	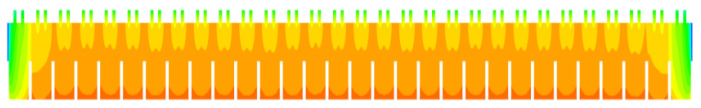
0.2	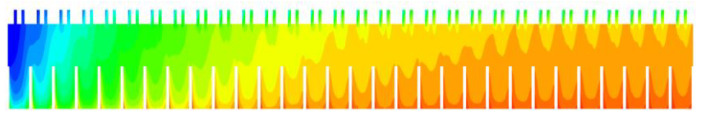
0.6	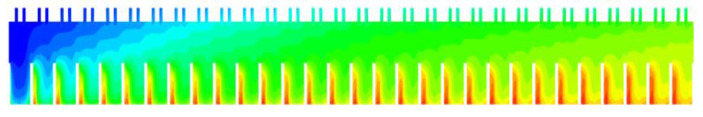
0.8	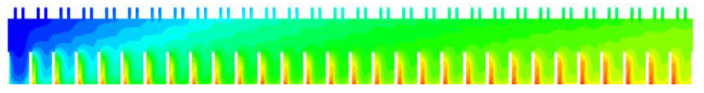
1.0	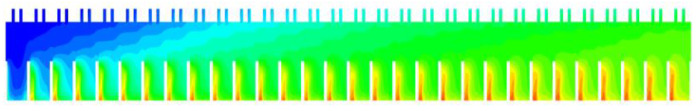
1.5	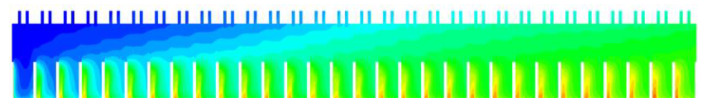
2.0	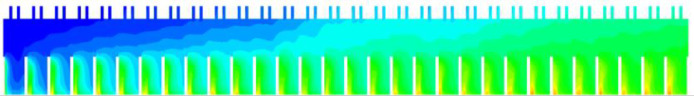

**Table 9 entropy-24-01550-t009:** Water vapor concentrations under different wind speeds with an evaporation space height of 16 mm (Model B).

Wind Speed(m/s)	Water Vapor Concentrations 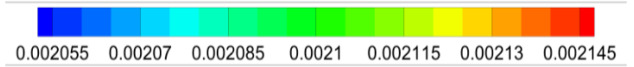
0	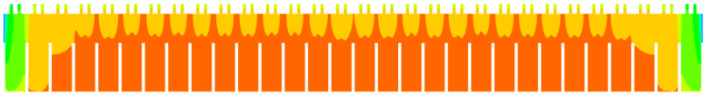
0.2	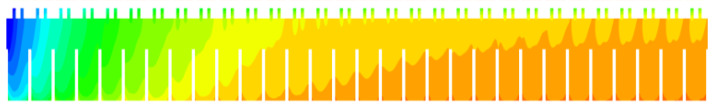
0.6	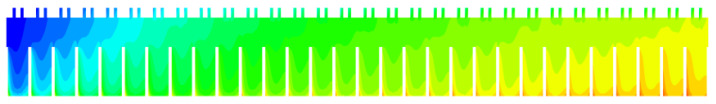
0.8	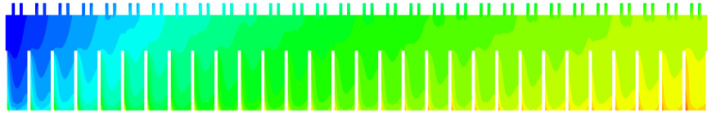
1.0	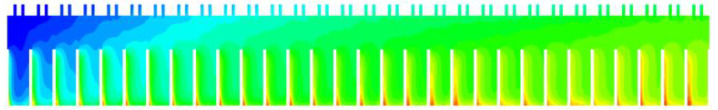
1.5	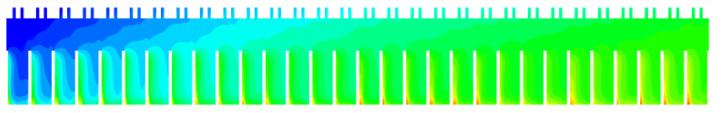
2.0	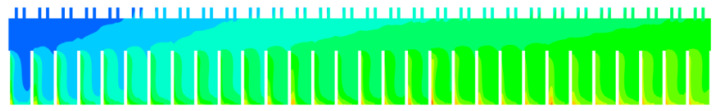

## Data Availability

Data are available on request.

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
