# Peer review of "Numerical Study on Effects of Wind Speed and Space Heights on Water Evaporating Performance of Water-Retained Bricks"

_entropy, 2022, doi:10.3390/e24111550_

Round 1

Reviewer 1 Report

The manuscript's title is excessively lengthy. If you can create a more compelling presentation with a shorter title, you should do so.

The English language should be thoroughly inspected, as should the document for grammatical errors.

Provide a list of abbreviations, units, and nomenclature underneath the keywords in the manuscript.

Please emphasize the strengths of the study.

Please clearly state the limitations of the study.

Pay attention to the use of tenses in citations in the introduction.

Page 2 line 47: ...showed that... or show that

Page 2 line 54, 64, 72, 77,79, 87 and so on.  It is necessary to use the present tense in words such as proposed (propose), showed that(show/shows that). The reason is that the articles have been published, and the author suggests in his article i.e., present tense.

Page 3 line 128: ...the authors proposed... or the authors propose

There are errors in your citation notation in paragraphs. You used et al in single-author and two-author articles.

Goudarzi et al. [8] compared ==== Goudarzi and Mostafaeipour

Sharifi et al. [12] systematically === Sharifi and Yamagata

Almodovar et al. [13] === Almodovar, and Roche,

Kharrufa et al. [14] conducted === Kharrufa, and Adil

Sarma et al. [15] theoreti === Sarma

Tang et al. [18] pro- === Tang and Etzion,

Pasquill et al. [19] studied === Pasquill

Vik et al. [20] === Vik, and Reif,

Soma et al. [24] measured === Soma and Kunugi,

Costa et al. [25] numerically === Costa,

Soma et al. [26] experimentally === Soma, and Kunugi,

Please arrengment the references section. Capital letters are used in some notations, such as [11]. 

Organize references according to the journal template.

Please increase the resolution of Figures 1 and 2.

Which paragraph refers to Figure 4? Please cite in-paragraph.

Which paragraph refers to Figure 7? Please cite in-paragraph.

Discuss results in concise and make the way for the future study which need to be addressed.

Conclusion section is missing some perspective related to the future research work.

Reviewer 2 Report

Referee Report for the manuscript “Numerical study on water evaporating performance of water-retained bricks under different wind speeds and evaporation space heights” by Rubing Han, Zhimao Xu and Enshen Long

This manuscript deals with the computational analysis of the water evaporating performance of particularly efficient bricks used in roof renovation methods, under different wind speeds and evaporation space heights. The paper is interesting for developing new energy-saving processes and the results can be used to improve the performances of these systems. However, many improvements are necessary to obtain an acceptable version of the manuscript with a sufficient scientific rigour. Since the paper deals with water-retained bricks the author should introduce the definition, the structure and the property of these objects at the beginning of their study, before any specific analysis. Their functional principle should also be made clearer and more explicit for reader not expert in the subject.
The acronym CFD probably means “Computational fluid dynamics” but it is not defined in the manuscript as it should be.
The difference between model B and model C is not clear and should better explained (in Fig.3, e.g., it is difficult to identify differences). What is the difference between “lid with upright hole to the air flow” and “lid with parallel hole to the air flow”? Please add explanations.
In the model definition, the positions of the x, y, z axes are not defined and it is therefore difficult to follow the structure of the model. Moreover, in the model outline, each equation should be complemented with the region where it is defined and used (with respect to the axes previously mentioned). Many quantities used in the equations are not defined at all. It is very important to define all variables and parameters (with units) and justify the proposed equations. The entire description of the equation system remains too vague and not scientifically acceptable. We remark that the description of the model must be accurate enough for another researcher to implement it in the exact same way. The whole article is written as a technical report for insiders, but a scientific article must be readable and understandable by any researcher. So please consider a complete revision of the presentation of the article that takes these comments into account.

Round 2

Reviewer 2 Report

The authors have properly replied to the questions and criticisms and therefore tha manuscript can be now accepted for publication as is.